# Enhancing an Oxidative “Trojan Horse” Action of Vitamin C with Arsenic Trioxide for Effective Suppression of KRAS-Mutant Cancers: A Promising Path at the Bedside

**DOI:** 10.3390/cells11213454

**Published:** 2022-11-01

**Authors:** Agata N. Burska, Bayansulu Ilyassova, Aruzhan Dildabek, Medina Khamijan, Dinara Begimbetova, Ferdinand Molnár, Dos D. Sarbassov

**Affiliations:** 1Department of Biology, Nazarbayev University, Astana 010000, Kazakhstan; 2National Laboratory Astana, Nazarbayev University, Astana 010000, Kazakhstan

**Keywords:** Kirsten rat sarcoma (KRAS) mutant ancers, Warburg effect, oxidative stress, arsenic trioxide (ATO), vitamin C (VC also known as ascorbic acid), reactive oxygen species (ROS), suicidal ROS production by mitochondrial (SRPM)

## Abstract

The turn-on mutations of the *KRAS* gene, coding a small GTPase coupling growth factor signaling, are contributing to nearly 25% of all human cancers, leading to highly malignant tumors with poor outcomes. Targeting of oncogenic KRAS remains a most challenging task in oncology. Recently, the specific G12C mutant KRAS inhibitors have been developed but with a limited clinical outcome because they acquire drug resistance. Alternatively, exploiting a metabolic breach of KRAS-mutant cancer cells related to a glucose-dependent sensitivity to oxidative stress is becoming a promising indirect cancer targeting approach. Here, we discuss the use of a vitamin C (VC) acting in high dose as an oxidative “Trojan horse” agent for KRAS-mutant cancer cells that can be potentiated with another oxidizing drug arsenic trioxide (ATO) to obtain a potent and selective cytotoxic impact. Moreover, we outline the advantages of VC’s non-natural enantiomer, *D*-VC, because of its distinctive pharmacokinetics and lower toxicity. Thus, the *D*-VC and ATO combination shows a promising path to treat KRAS-mutant cancers in clinical settings.

## 1. Introduction

### 1.1. KRAS Is a Crucial Component of Growth Factor Signaling

The *Kras* gene was originally identified in the Kirsten rat sarcoma (Kras) virus DNA sequence, and its transforming oncogenic potential was described in the early 1980s [1,2]. RAS proteins are recruited from cytoplasm to the plasma membrane signaling nodes upon activation of a tyrosine kinase receptor where, together with various regulatory proteins, it activates signaling pathways coordinating cell growth and proliferation. A RAS protein translocation to the cell surface happens via one of two possible routes involving certain post-translational modifications, such as farnesylation of the cysteine residue of the RAS CAAX motif and α-carboxyl group methylation (Figure 1a) [3]. RAS proteins belong to small G protein groups and are transitioning between guanosine triphosphate (GTP)-bound active and guanosine diphosphate (GDP)-bound inactive forms [4,5]. Activation happens through exchange of GDP by GTP guanine nucleotides and is closely regulated by guanine exchange factors (GEFs) (Figure 1a) [6,7]. RAS activation (a GTP bound state) changes the protein conformation that allows its interactions with more than 20 different proteins from 10 effector families [4].

RAS triggers activation of the critical RAF/MEK/ERK (MAPK) kinase (Figure 2 left) and PI3K/AKT/mTOR signaling pathways (Figure 2 right) [5]. Signaling pathways are activated by a ligand binding with a specific receptor, such as the epidermal growth factor (EGF) with its receptor tyrosine kinase, EGFR, leading to its activation, mediated by the EGFR dimerisation and cross-phosphorylation of each other on tyrosine residues (also known as a ligand-dependent autophosphorylation). A specific tyrosine phosphorylated site on EGFR recruits growth factor receptor-bound protein 2 (GRB2) by its SH2 domain which serves as the adaptor protein by further recruiting the son of sevenless (SOS) and RAS proteins. An SOS protein is the guanosine nucleotide exchange factor (GEF) that serves as the RAS activator by catalyzing its GDP exchange to GTP and turning on the cascade of kinases starting from RAF to MEK and finally to ERK also known as MAPK (Figure 1c). The ERK kinases activate transcription factors promoting cell growth and proliferation [5]. The RAS family of proto-oncogenes comprises HRAS, KRAS, and NRAS, which are among the most mutated genes in human cancers (Figure 1a). These genes are ubiquitously expressed and share a significant sequence homology and mainly overlapping roles.

### 1.2. KRAS Is a Potent Oncogenic Driver

Recent statistical data indicate that approximately 25% of cases of cancer carry RAS mutations. Analysis of combined datasets from four leading cancer mutation databases detected KRAS in 19 out of 29 cancer types [8,9,10] and was marked as the “Everest” of oncogenes [11]. KRAS is the most frequently mutated isoform among the RAS family of genes and is associated with an overall percentage of 75–83% of all RAS-mutated cancers, whereas NRAS is detected in only 8–17% and HRAS in 3–7% of all cases. Human pancreatic ductal adenocarcinomas (PDAC) carry the highest frequency of KRAS mutations (about 95%) whereas in colon and lung cancers it is approximately 50% and 35%, respectively [10,12,13]. A high KRAS mutation load is detected in other types of cancers, such as multiple myeloma, ovarian, uterine, and stomach cancers (22%, 15%, 18%, and 16%, respectively), whereas NRAS mutations are frequently detected in melanoma (30%), multiple myeloma (18%), acute myelogenous leukemia, colorectal cancer (CRC) (10%), and thyroid cancer (8%), and HRAS is the least frequently mutated RAS gene detected in cancers of the bladder, head, and neck squamous cell carcinomas and the uterus (5% or less) [12]. These statistics largely vary due to the biological variation and populations used in different studies as well as the sampling bias within datasets. Among all mutations in KRAS and NRAS isoforms, the most dominant hotspots are located in codons G12, G13, and Q61 and comprise 96%–98% of all point mutations [14,15,16]. In the KRAS gene, about 81% mutations happen at position G12, and 14% at G13, but very few (2%) at codon Q61. On the contrary, 62% of NRAS tumor mutations are located in codon Q61, 23% in codon G12, and 11% G13. In HRAS mutant tumors, approximately 26% of mutations take place on the codon G12, 23% on G13 and 38% on Q61 [17]. Within 100 mutation sites known in all three RAS isoforms, the most common mutant alleles are located on G12 site in KRAS with 6 point mutations overall. The most common KRAS G12 mutations are identified as G12D (42%), G12V (28%), and G12C (14%), whereas G12A, G12R, and G12S are much less frequent [18,19]. A distribution of point mutations in KRAS also varies substantially in tumor tissues; for example in pancreas probability of the G12R mutation is 13% but it is only up to 2% in intestine. Moreover, different mutations alter RAS signaling with distinctive features as has been reported for the KRAS G12 and G13 mutants [19]. The KRAS mutations on Q61 site also promote change in morphology, growth transformation, signaling, and metabolism of cells [20]. It has been shown that different KRAS mutant alleles may have different clinical impacts on the prognosis of pancreatic and CRCs [21,22,23,24]. KRAS cancers are a typical example of oncogene “addiction” also known as KRAS dependency. Oncogene addiction is a phenomenon that arises when expression and activity of a single abnormally activated gene is necessary to be sustained despite the accumulation of other multiple oncogenic mutations [25]. A KRAS-dependent gene expression signature composed of 46 differentially regulated genes was able to correctly identify 15 out of 18 cell lines bearing KRAS mutations from non-KRAS mutations. This signature also allowed accurate prediction of KRAS dependency across well-differentiated KRAS human mutant cancer samples with a low misclassification error [26]. It has been concluded that a KRAS dependency is associated with epithelial differentiation status and its mechanism is epigenetically imprinted [26,27]. The other RAS Dependency Index (RDI) is computationally derived from a single sample gene set enrichment analysis (ssGSEA) by using an expression of sets of genes. A strong connection between RDI and patient survival rates and its potential clinical utility was described [28]. However, it needs further validation studies on larger patient cohorts.

### 1.3. Challenges of KRAS Targeting

The KRAS-driven cancers represent highly malignant oncologic disorders with a poor clinical outcome. A specific and potent targeting of this highly malignant oncogenic pathway is one of the most challenging and demanding tasks in oncology. Therapies already in use for treatment of KRAS-mutant cancers such as cetuximab, panitumumab, gefitinib and erlotinib present a big challenge in terms of the way cancers overcome the treatments and develop resistance [29]. Panitumumab and cetuximab are humanized anti-epidermal growth factor receptor (EGFR) monoclonal antibodies, which are more successful in patients with the wild-type rather than mutant KRAS tumors [30]. A retrospective overview showed that the cetuximab treatment is ineffective in cancers with KRAS mutations, and testing of KRAS genotype as a predictive biomarker is necessary for cetuximab or panitumumab therapy in CRC [31]. In past decades, there were no effective therapies available for KRAS cancers and they were “undruggable” because KRAS has no hydrophobic pockets for its inhibition [32,33]. However, recent years have brought considerable progress and several KRAS-mutant-specific inhibitors were developed such as ARS-853, AMG 510, MRTX849, and MRTX1133 (Figure 3) [34]. AMG 510 (generic name lumakras and sotorasib) was approved by the FDA in 2021 for treatment of a KRAS G12C mutation-positive non-small cell lung cancer (NSCLC). In silico drug discovery methods employing drug–gene network analysis, which screen the FDA-approved drug library, can lead to repurposed inhibitors of oncogenic KRAS. Decitabine is one such drug and was shown to be a potent growth inhibitor of KRAS-dependent pancreatic cancer cells and in patient-derived xenograft models. It uses pyrimidine biosynthesis as a metabolic vulnerability for KRAS-dependent PDAC [35]. Advances in drug development and formulation are promising in targeting highly malignant KRAS-mutant tumors. However, a dynamic development of cancer drug resistance remains a challenging problem in oncology.

## 2. Targeting of the Specific KRAS-Mutants

Recent developments, however, attempted to overcome a shortcoming in KRAS targeting and the specific compounds with novel mechanisms of action were developed to a specific KRAS G12C mutant (Table 1). Novel inhibitors covalently target the G12C mutation in the switch II pocket of the protein, which blocks binding of the mutant KRAS with GTP (preventing effector interactions) and halts downstream signaling pathways, such as RAF/MEK/ERK [36].

### 2.1. Every Common Specific KRAS-Mutant: A Case for the Drug Development

Structurally, KRAS consists of an effector binding lobe, comprising the first 86 residues, an allosteric lobe, and a carboxy-terminal region (Figure 1a). The effector lobe has the phosphate-binding loop within a stretch of amino acid residues 10–17 (P-loop), the switch I between amino acid residues 30–38, and the II loop spans between amino acid residues 60–76. The loops of the switch I and II, mediate the protein–protein interactions with downstream signaling effector protein kinase RAF1. The GTP hydrolysis activity of KRAS is performed by the Q61 site. The KRAS mutations at positions G12, G13, Q61, and A146 are located at the effector lobe and lead to a shift toward the active KRAS form through impairing nucleotide hydrolysis and/or activating nucleotide exchange [48]. Shokat and colleagues were the first to find a hidden pocket to target KRAS and effectively use it in therapy. This pocket is located in the switch II next to the G12C mutated site (Figure 1b right) [36]. A steric distance between the switch II and mutated C12 site made possible the development of covalent inhibitors of the switch II, thereby achieving allosteric inhibition of cysteine in G12C to prevent the nucleotide exchange catalyzed by GEFs and diminished the subsequent interaction between RAS and RAF [49]. The area of a covalent inhibition is pointed in the switch pocket.

The improved version of this drug, ARS-1620, demonstrated the tumor growth control in a KRAS G12C xenograft model [49] by its additional H-bond interaction with H95 in the switch pocket and a more favorable alignment of the warhead toward C12 (Figure 4c) [50].

There is also a different compound MRTX849 (Adagrasib) designed to target this KRAS G12C mutation. It acts by forming a hydrogen bond (H-bond) with D69, a salt bridge with E62 and a cation-π interaction with H95 in the switch pocket (Figure 4b) [51]. MRTX849 showed a 45% response rate and 8.2-month median duration of response in non-small lung cancer tumors [52]. A different mode of action is carried out by AMG510 (sorotasib); it does not form a H-bond with H95, but instead induces a conformational change of the histidine, resulting in the formation of a new cryptic sub-pocket formed by H95, Y96, and Q99 (Figure 4c). The compound demonstrated a 37.1% response rate, a progression-free survival of 6.8 months and a median overall survival of 12.5 months in a phase II clinical trial of 126 patients with advanced non-small-cell lung cancer [53]. MRTX1133 is a selective non-covalent inhibitor of KRAS G12D because the aspartate in KRAS G12D has a carboxyl group that is less nucleophilic than the sulfhydryl group of cysteine; therefore, it does not work for G12C (Figure 4d). MRTX1133 binds to the switch II pocket and inhibits the protein–protein interactions necessary for the activation of the downstream pathway of KRAS [44]. Other inhibitors, ARS-853 and AMG 510, target specifically KRAS G12C mutation and do not act on the KRAS G12V or G12D mutants occurring more frequently in human cancers [49,54].

The novel therapeutic development targeting KRAS-mutant cancers also works on discovery of pan-Kras inhibitors and proteolysis targeting chimeras (PROTAC) which avoid HRAS and NRAS [55,56]. Various inhibitors targeting upstream and downstream RAS signaling pathways were reviewed in [56,57]. Promising results of the pre-clinical studies using combinations of inhibitors simultaneously targeting several KRAS signaling regulators including SOS1, SHP2, and EGFR were also reported [58,59,60,61]. The phase I and IIb clinical trials of RMC-4630 (SHP2-inhibitor) plus LY3214996 (ERK-inhibitor) in patients with KRAS mutations in CRC, NSCLC, or PDAC are in progress (NCT04916236). Many other SHP2 inhibitors (TNO155, JAB-3068, JAB-3312, RLY-1971, BBP-398, ERAS-601, PF-07284892/ARRY-558, and SH3809) and SOS1 (BI 1701963, RMC-5845, BAY-293, SDGR5, GH52, ERAS-9, and SOS1i) are in the clinical and preclinical phases [56]. A potentially effective strategy is also to inhibit mutated KRAS gene expression at the mRNA level. A significant reduction in tumor size (pancreatic cancer xenografts) in mice was observed after CRISPR-Cas13a-mediated KRAS G12D mRNA knockdown. A specific KRAS mRNA silencing induced a potent apoptosis in in vitro models [62]. The RNA interference approach to silence KRAS oncogene expression was also tested [63,64]. Novel approaches such as siRNA encapsulated in exosomes and mRNA vaccines are in the clinical trials. Recent pre-clinical studies showed that fibroblast-derived exosomes loaded with G12D siRNA (iExosomes) efficiently attenuate PDAC tumor growth and extend tumor bearing mouse survival [65]. Two of KRAS mRNA vaccines are in phase I clinical trials: mRNA-5671/V941 co-developed by Moderna and Merck (NCT03948763) and ELI-002 by Elicio Therapeutics (for KRAS/NRAS-mutant solid tumor NCT04853017). Lipid nanoparticles encapsulated mRNA vaccines encoding mutant KRAS epitopes G12D, G12V, G13D, and G12C are taken up and translated in antigen-presenting cells, and then presented by MHC molecules on a cell surface [66].

Although some of the described novel therapies show promising results, it is evident that the drug resistance is a common feature of KRAS-mutant cancers that will be a most challenging task to overcome, making it a critical limiting factor in eradication of highly malignant cancers [67]. Therefore, a combination of the cancer targeting therapies with the focus to obtain a potent killing of KRAS-mutant cancer cells will be most effective, if a synergistic potentiation in the drug action outcomes can be realized.

### 2.2. How Resistance Gets Evolved to the Specific KRAS G12C Mutant Inhibitors

A potent targeting of specific KRAS G12C mutations with a high precision results in the further mutations that disrupt covalent or potentially non-covalent drug-binding and leads to drug resistance. The KRAS R68S and Y96C mutations located within the switch II pocket of the MRTX849 and AMG510-binding site interfere with the drug non-covalent binding interactions and decreases binding affinity substantially [68]. Another KRAS mutation within the same Y96D site alters the switch II pocket and reduces the H-bonding between the Y96 residue of KRAS and MRTX849 [69].

Alternatively, other KRAS mutations such as G13D, A59S, K117N, and A146P residing outside the drug-binding pocket are altering the functional activity of KRAS by enhancing its nucleotide exchange and maintaining its active GTP-bound state not accessible for the drug binding [68]. It has been also indicated that a conformation-specific KRAS G12C inhibitor leads to a rapid non-uniform adaptation where some cancer cells stabilize its active drug-insensitive state and maintain cancer cells in a drug-resistant state [70].

Various studies revealed that KRAS inhibition is possible to suppress by a feedback activation of upstream or downstream mediators and other negative regulators. Cells activated with EGF showed reactivation of KRAS in the cells, suggesting that EGFR mediates resistance to KRAS G12C inhibitors [71]. The changing activity of the downstream effectors of KRAS also leads to occurrence of drug resistance. For example, a sub-clonal evolution of MET amplification in KRAS G12C non-small cell lung cancer cells that have become resistant to AMG510 in vitro [72]. Another way of inducing resistance for AMG510 or MRTX849 might be the continued FAK activation, which leads to weak treatment outcomes by dysregulating FAK-YAP signaling. Focal adhesion kinase (FAK) is a non-receptor kinase which plays a role in cell growth regulation and transduction of signals [73]. Moreover, cells which acquired resistance to AMG510 seemed to undergo epithelial-to-mesenchymal transition (EMT). Such a kind of induction leads to activation of expression of several RTKs, such as ERBB3 and FGFR. Moreover, drug resistance induced by EMT causes the enhancement of PI3K/AKT and MAPK signaling [74]. Another mechanism that involves the development of so-called adaptive resistance, whereby KRAS-dependent tumors under the treatment pressure converse from one histological type to another, is driven by the primary cancer type. The transformation from adenocarcinoma to squamous cell carcinoma was observed during MRTX849 treatment without any other drug-resistance mechanisms [75]. Hallin et al. showed that a cell cycle dysregulation changed KRAS-mutant allele frequency, which is an additional factor that could suppress the therapeutic response to MRTX849 [38]. Hereby, we described only a few possible mechanisms of resistance toward novel KRAS-specific inhibitors with other existing and with the potential of KRAS-mutants to develop new complex ways to disable treatment efficacy. To overcome these, the different combinations of drugs are applied to block upstream, downstream, and other factors of regulation which are associated with KRAS signaling.

## 3. Deregulation of Metabolism in Cancer

The main source of energy and ATP production in primary cells is mitochondrial oxidative phosphorylation (OXPHOS), whereas in tumor cells the production of ATP by OXPHOS is low and most of the glucose is utilized by its fermentation to lactate even in the presence of oxygen, the process known as an aerobic glycolysis or Warburg effect (Figure 5) [76,77,78]. In normal cells, the pyruvate from glycolysis preferentially enters the mitochondrial matrix where it is oxidized to acetyl-coenzyme A (acetyl-CoA) by the pyruvate dehydrogenase (PDH) complex. Acetyl-CoA is then metabolized through tricarboxylic acid (TCA) cycle, followed by OXPHOS for a productive ATP generation (up to 38 ATP molecules per one molecule of glucose are generated) [79]. An abnormal glucose metabolism is inherent in cancer cells as one of the well-studied characteristics first described by Warburg who reported association between an elevated glucose uptake and decrease in oxygen consumption with an increased production of lactic acid in aerobiosis. In cancer cells, the majority of produced pyruvate is uncoupled from the mitochondrial TCA cycle and OXPHOS processes, and is converted by lactate dehydrogenase (LDH) to lactate. Accumulated lactate facilitates tumor progression and acidification of the tumor environment, which in turn promotes tumor proliferation, metastasis, and resistance to developed antitumor therapies [80].

### 3.1. KRAS Cancer Mutations and Metabolic Reprogramming

An oncogenic KRAS sets the metabolic changes by hyperactivation of growth factor signaling that ultimately leads to increase of a cellular glycolytic flux facilitating tumor growth [81,82]. Intensive anabolic processes in cancer cells are maintained by an enhanced glucose consumption and its fermentation to lactate (aerobic glycolysis) [83]. It generates an anaerobic ATP in cancer cells supporting growth in hypoxic conditions of a tumor environment. The advantage of a high glycolytic flux is a continuous supply of glycolytic intermediates for the pentose phosphate pathway to gain an elevated synthesis of nucleotides and phospholipids for actively growing and proliferating cancer cells, which is highly relevant for KRAS-mutant cancers because of their glucose addiction.

The metabolic shift in KRAS-mutant cancer cells relies on a high absorption of glucose mediated by abundant glucose transporters (GLUT) where GLUT1 is considered as a key supplier of glucose in cancer cells promoting an aerobic glycolysis in tumorigenesis (Figure 5) [84]. Glucose metabolism intermediates are then channeled into non-canonical biosynthesis pathways such as the non-oxidative arm of a pentose phosphate pathway to produce ribose for biosynthesis of RNA/DNA and the hexosamine biosynthesis pathway (HBP) that produces glycosylation precursors [85,86,87]. A high reliance on glutamine of cancer cells for tumor growth was also reported, whereby the glutamine-fueled TCA cycle results in generation of ATP, reactive oxygen species (ROS), NADPH, amino acids, nucleotides, and lipids that are critical for KRAS oncogene-induced tumorigenicity [85,88,89,90,91].

### 3.2. A Basal Oxidative Stress Is a Liability of Malignant KRAS-Mutant Cancers

An altered cellular signaling increases the production of ROS as well as activating antioxidant programs, which are advantageous and drive tumorigenesis of KRAS-mutated cancers. An oncogenic KRAS can promote ROS generation by regulating HIFs, altering mitochondrial function by suppressing the respiratory chain complex I and III and inducing expression of the transferrin receptor. To balance the redox state, KRAS-mutant cancer cells display upregulation of major antioxidant enzymes (peroxiredoxin 3, thioredoxin peroxidase, catalase) and reduced glutathione (GSH) levels, as well as enhanced detoxification pathways and resistance to apoptotic death in response to oxidative compounds such as hydrogen peroxide (H_2_O_2_) [84]. It is still not clear how the RAS oncogene modulates the redox equilibrium or whether pro- or anti-oxidant elements contribute to a RAS-induced transformation and tumorigenicity. Expression of RAS itself can induce ROS production in cancer as a result of increased metabolic rate, genetic mutations, and hypoxia. A mitochondrial ROS generation is essential for KRAS-mediated cancerogenesis [92]. Apart from mitochondrial ROS, NADPH oxidase 1 (NOX1) increased expression and activity have been observed in KRAS-driven CRCs leading to additional ROS production. A NOX1-mediated ROS production is essential to support RAS transformation [93,94].

The translocalization of KRAS protein from the plasma membrane to the mitochondria was observed in KRAS G12V-expressing cells by a doxycycline inducible system. The expression of KRAS G12V protein causes a dysfunction of mitochondria by a significant decrease in mitochondrial respiratory activity (complex I), oxygen consumption, and mitochondrial membrane potential ΔΨm. In addition, an elevated ROS generation and adaptive upregulation of GSH synthesis in response to the sustained ROS production have been observed in this cellular model [95]. A KRAS hyperactivation alters a mitochondrial morphology caused by transformations in mitochondrial fusion and fission with subsequent changes in cellular metabolism [96]. However, oxidative stress is a vulnerable state of KRAS-mutant cells and can be used to induce cytotoxic response. A glucose withdrawal leads to increased oxidative stress driven by NOX1 and mitochondria and can activate a number of kinases such as ERK, JNK, and LYN in cells carrying KRAS mutations which are dependent on glucose for survival [97]. Glucose deprivation activates an amplification loop, with an increased phospho-tyrosine signaling, until ROS accumulates above a toxicity threshold. Once it exceeds the capacity of cell antioxidant systems, it triggers a cell death.

## 4. The Vitamin C Cancer Treatment Is Back: How a Famous Antioxidant Turns to a “Trojan Horse” Oxidant in KRAS-Mutant Cancer Cells

### 4.1. Vitamin C (Ascorbic Acid) in Cancer Treatment

Vitamin C (VC) or ascorbic acid is a water-soluble vitamin and antioxidant present in many vegetables and fruits. It plays an essential role in the majority of cellular processes. An increased risk of cancer mortality is associated with VC deficiency [98,99]. In high doses, VC is almost non-toxic to normal cells in in vitro experiments but highly toxic for a number of human tumor cell lines. The pioneering study by Linus Pauling and Ewan Cameron (double Nobel Prize winners) showed anticancer abilities of high doses of VC in the 1970s. The end-stage cancer patients were treated with 10 g per day of VC combined with chemotherapy, and their survival rate reached above one year [100]. The presence of KRAS mutations in cultured human CRC cells was associated with selective killing by high levels of oxidized VC, suggesting that KRAS status can at least partially explain response to VC [101]. Recent in vitro and in vivo preclinical studies of VC in various human cancers confirmed its efficacy and cytotoxic effects on cancer cells [102], but efficacy in clinical trials was limited [103,104].

Administration of VC to patients by an intravenous (IV) route is the most effective way to deliver a high dose of the drug. It appeared that an IV administration of VC achieved millimolar levels of the drug sufficient to provoke cytotoxicity in cancer cells [105]. In a phase I clinical study, the IV infusion of 100 g of VC reached 25–30 mM level of the drug in circulation that decreased only to a 10-mM range following 4 h. Such a high dose of VC displays antitumor activity sufficient to slow growth of cancer cells [106]. However, the oral administration of VC reached only 200 μM with a further increase to 400 μM by a special encapsulated formulation of VC [105,107]. Novel VC pharmaceutical formulations focus on increasing structural stability and cellular uptake [108,109].

Numerous studies demonstrated potential anticancer activity of VC alone or in combination of a high dose of VC with other conventional therapies. First, a human phase I clinical trial was conducted, with VC administration by IV with the radiotherapy enhanced radiosensitization of pancreatic cancer, while a protective effect from radiation was observed in a surrounding tissue [110]. It has been also reported that combining of a high dose of VC with the anti-cancer drugs eribulin mesylate, tamoxifen, fulvestrant, or trastuzumab was cytotoxic for breast cancer cells [111]. The efficacy of an immune checkpoint therapy (ICT), targeting PD-1/PD-L1 and CTLA-4 molecules in breast cancer or CRC cell lines was enhanced by the co-treatment with VC [112]. Other studies show that VC at 50 μM can reduce viability of HCT116 cells when combined with the anti-neoplastic DNA-demethylating agents decitabine (DAC) and azacytidine (AZA) [113]. An intravenous administration of VC at 25–50 g per day every 1–2 weeks in patients with small cell lung cancer (SCLC) in combination with the alkali therapy results in median survival increased to 44.2 months compared to the control group with 17.7 month survival [114].

Calorie-restricting diets (low glucose or proteins) are referred to as fasting-mimicking diets (FMDs). The anti-tumor additive or synergistic effects were shown following the co-treatment of FMD with chemotherapy or number of targeted therapies in several preclinical breast, lung, and colorectal cancer models [115]. FMD contributes to sensitizing cancer cells to chemotherapy by increasing ROS production [116,117]. Importantly, it has been shown that FMD can increase anticancer activity of VC in KRAS-mutant CRCs. The synergistic effects of FMD and VC treatments were observed in cell culture and animal models in vivo. The combination of FMD and VC was the most effective and also most well-tolerated, with low toxicity [118]. FMD enhances a VC’s cytotoxicity in KRAS-mutant cancer cells by increasing reduced ferritin (a protein which binds iron), which further leads to an increase in reactive iron level, ROS production, and cell apoptosis. This effect can be further boosted by chemotherapy [118]. This promising combination should be further tested by well-designed randomized clinical trials in CRC and other KRAS-mutant tumors.

### 4.2. The Anti-Cancer Action of VC: How Does It Work

Metabolic reprogramming sensitizes KRAS-mutant cancer cells to a high dose of VC by meddling in an epigenetic regulation, hypoxia signaling, and intracellular iron metabolism (Figure 6a) [119]. The high dose of VC kills cancer cells by provoking an oxidative stress in highly malignant cancer cells including KRAS-mutant cancer cells, but how does a potent antioxidant VC become an oxidative factor in cancer cells? It is highly evident that a pro-oxidant action of VC depends on metal-ion redox chemistry. In particular, free iron was shown to be essential for VC-induced cytotoxicity [120]. The oxidative impact is carried out by the oxidized form of VC known as dehydroascorbate (DHA) and the distinct locations of the oxidation VC to the DHA reaction and by reducing DHA back to VC reaction to determine the oxidation of cancer cells. VC gets oxidized in a body to DHA. KRAS mutant cancer cells absorb intensively DHA by a glucose transporter GLUT1 because DHA structure resembles glucose. KRAS metabolic reprogramming associated with elevated expression of GLUT1 allows for an increased uptake of DHA to the cytoplasm, which leads to a disruption in the Warburg effect [120,121]. DHA inside a cancer cell is reduced back to VC at the cost of GSH oxidation. In a high scale, an active absorption of DHA, a decoyed form of glucose, results in a substantial exhaustion of a cellular GSH leading to oxidative stress coupled to ROS accumulation and inactivation of glyceraldehyde 3-phosphate dehydrogenase (GAPDH). A continuous inhibition of GAPDH activity induces oxidative damage via auto-oxidation, which is cytotoxic for highly glycolytic KRAS-mutant cancer cells [101,102].

Elevation of ROS to cytotoxic levels provokes irreversible damage to DNA and mitochondria and triggers apoptotic pathways in cancer cells [122,123,124,125]. In highly glycolytic KRAS-mutant cells, an accumulation of ROS provokes an apoptotic cell death via inhibition of GAPDH, a glycolytic enzyme that has an ability to reduce adenosine triphosphate (ATP) leading to energetic collapse and cell death which is not observed in KRAS wild type cells [101]. A highly glycolytic metabolism of KRAS-mutant cancer cells can be also disrupted by VC through downregulation of important metabolic checkpoints such as GLUT1 and PKM2 [121].

A redox damaging of VC also interferes with cell signaling and mitochondrial function. The effects of VC on EGFR/MAPK signaling pathway in KRAS-mutant CRC lines has been reported, indicating inhibition of MEK1 and ERK1/2 phosphorylation without changes in their protein expression [121]. VC alters a mitochondrial function in KRAS-mutant cancer cells. The VC-dependent modulation of pyruvate dehydrogenase (PDH) activity, as well as a TCA cycle, was reported. In this study, VC induced a significant ATP depletion, quick dissipation of mitochondrial membrane potential and decreased phosphorylation of PDH’s component E1-α on S293 site leading to an increase in PDH, citrate synthase activity, and also an increase in the conversion of pyruvate into acetyl-CoA. Moreover, the downregulation of PDK-1 has been linked to a VC-mediated hydroxylation of proline (P402) in hypoxia inducible factor-1α (HIF-1α) [126]. It is further supported by finding that the VC treatment of pancreatic adenocarcinoma cells suppressed HIF-1α, which is known to contribute to tumorigenesis. The effect was observed through activation of HIF hydroxylases, which in turn inhibits transcription of HIFs, resulting in delays in tumor growth [127]. It was also reported that high doses of VC are effective in suppressing Apc/Kras G12D mutant tumors [101]. Thus, VC is a potent suppressor of KRAS-mutant tumor growth by provoking an oxidative stress and meddling with cancer metabolism.

## 5. A Mild Oxidant ATO Is the Cancer Drug: The Mechanism of Action

### 5.1. The History of ATO: Harnessing the Poison for Cancer Treatment

An inorganic arsenic has long been used as the active ingredient in a traditional Chinese medicine for treatment of numerous conditions including tumors for over 2000 years [128]. In the early 17th century, arsenic trioxide (ATO) solution in potassium bicarbonate was developed by Thomas Fowler and used to treat diseases such as asthma, chorea, eczema, pemphigus, and psoriasis and was later used also to treat anemia, Hodgkin’s disease, and leukemia. Suppression of leukocytes by ATO was reported in 1878 [129,130,131] and later, a treatment of leukemias with ATO alone and in the early 20th century, combined with radiotherapy, was used [132]. For acute promyelocytic leukemia (APL), treatment with ATO was used for the first time in the early 1990s, when it showed high a clinical remission rate. An injectable formulation of ATO, known as trisenox, was approved in 2000 by the FDA for treating relapsed or refractory APLs. The majority of patients with APL achieve a complete remission [133,134]. The ATO treatment was not well-suited for solid tumors due to a dose-related toxicity and unfavorable pharmacokinetics. However, some promising results of ATO treatment have been observed in patients with refractory/relapsed CRC resistant to 5-flurouracil, which is the most widely used clinical treatment [135,136].

### 5.2. The Oxidizing Impact of ATO

Arsenic oxidants execute the substantial stress-related cellular changes by inducing apoptosis, halting proliferation, stimulating differentiation, and inhibiting angiogenesis through various mechanisms. The main mechanism of an antitumor ATO’s action is induction of an ROS-dependent apoptosis (Figure 6b). Exposure to ATO provokes an oxidant–antioxidant imbalance in cells, as it can promote generation of variety of ROS, such as H_2_O_2_, superoxide (O2¯•), nitric oxide (•NO), perhydroxyl (HOO•), dimethyl arsenic peroxyl ((CH_3_)_2_AsOO•), and dimethyl arsenic ((CH_3_)_2_As•) radicals causing oxidative stress. Accumulation of intracellular ROS alters mitochondrial membrane potential causing its dissipation, release of cytochrome c and activation of the caspases, ultimately inducing the mitochondrial apoptotic pathway [137]. The cytotoxic drugs including ATO often target mitochondria, which then initiate apoptotic processes. Arsenic is recognized as an uncoupler of mitochondrial oxidative phosphorylation. This effect is achieved by the ATO binding to reactive thiol groups on proteins (ligands) located on cysteine residues. Cysteine residues are critical to many unique mitochondrial functions, including ROS production. Mitochondrial proteins, including cysteine residues in these proteins, undergo redox changes through modifications to the thiol groups on cysteines. The localization of proteins and their interactions are greatly influenced by oxidative post-translational modifications (PTM) of cysteine residues, which in turn are ROS-dependent. These cysteine PTMs include oxidation to sulfenic/sulfinic/sulfonic acids, *S*-nitrosation, *S*-glutathionylation, and disulfide-bond formation [138]. An ATO binding to thiol groups of mitochondrial enzymes involved in electron transfer chain results in impaired tissue respiration. A direct binding of ATO to proteins results in thiol oxidation leading to changes of various protein conformation and to inhibition of their function.

Glutathione (GSH) is a non-protein thiol and a reducing substance essential for the detoxification of endogenous and exogenous oxidative compounds. It makes it an important element in the antioxidative redox system of cells, counterbalancing ROS produced by cellular metabolism or invading oxidants from outside. GSH directly neutralizes ATO by binding arsenic, and it also controls its metabolism by facilitating a cellular arsenic uptake, alteration of arsenic methylation and stimulation of excretion of methylated arsenic forms [139,140]. Arsenic can decrease GSH levels in three ways: first, by reducing arsenates to arsenites using electrons donated by GSH; secondly, by direct binding thiol groups of GSH; and thirdly, by inducing ROS which cause GSH oxidation [141]. Normal and cancer cells with reduced GSH levels are highly sensitive to ATO treatment [142]. In fibroblasts and APL cells, ATO dysregulates a redox balance by increasing intracellular ROS levels and simultaneous depletion of GSH [143]. It has been clearly demonstrated that cancer cells with lower levels of glutathione peroxidase and catalase are highly sensitive to arsenite. Moreover, the experimental conditions leading to lowering cellular GSH levels sensitize cells to arsenite with natural or acquired resistance to the oxidant, while cells with greater levels of reduced GSH are protected from an ATO-induced apoptosis [143,144].

### 5.3. The Action of ATO on a Cellular Metabolism and Its Nuclear Effects

ATO inhibits glycolysis by direct binding to the cysteine residues (C256 and C704) on hexokinase 2 (HK2) and halting its activity [145]. Pyruvate kinase (PKM2), another known target of ATO, can modulate oncogenic anaerobic glycolysis, known as the Warburg effect, inducing further tumorigenesis and proliferation of cancer cells because PKM2 deletion can enhance oxidative phosphorylation [146]. Moreover, the HK2 overexpression significantly rescues the cells from ATO-induced apoptosis [145,147,148]. Exposure to inorganic arsenic can significantly inhibit many other enzymes involved in glucose metabolism and further induce insulin resistance and diabetes [141]. ATO can also activate caspases, which can directly trigger many apoptotic molecular and structural changes [144,149,150,151] as well as can promote non-classical apoptosis pathways such as ETosis, extracellular DNA traps, and pyroptosis [152]. Moreover, an inorganic arsenic acts as an epigenetic modifier of genes involved in the critical cellular processes, such as cellular growth and immune response [153]. It can induce substantial epigenetic changes by the inhibition of DNA methyltransferases [154]. In CRC cells, ATO altered transcriptional activity of several unmethylated cell cycle regulatory genes including cyclin B1, E1, D1, and GADD45A, which causes cell cycle arrest [155]. The antitumor action of ATO can be also facilitated by up- or downregulation of miRNAs in cancer cells and consequently decreased growth, increased apoptosis, or suppression of cancer migration and invasion [156]. Moreover, downregulation of certain miRNAs can sensitize cancer cells to arsenic treatment [157,158]. Studies have shown that ATO inhibits the nuclear factor (NF)-κB signaling pathway in leukemia cells and hepatocellular carcinoma stem cells by downregulating expression of p65, p50, p52, P65, c-rel, and RELB members of the NF-κB pathway [159,160]. The diverse action of ATO highlights a necessity of further mechanistic studies to reveal its anticancer effects. The downside of ATO’s therapeutic effects is that it triggers severe side effects including cardio-toxicity, hepatotoxicity, neurotoxicity, and skin problems and affects kidney function, which limits its clinical applications. Combinations of ATO with other FDA-approved drugs have been tested, showing better results with less adverse effects.

## 6. ATO Potentiates an Oxidative Effect of VC and the Combination of Both Drugs Is Effectively Killing KRAS-Mutant Cancer Cells

The development of potent therapeutics targeting oncogenic KRAS signaling remains a most critical and challenging area in oncology. Although specific KRAS-mutant cancer inhibitors were developed recently with a hope of resolving the critical problem in cancer treatment, their applications gain only a partial impact because tumor resistance occurs at a high rate. It is likely that the targeting of specific KRAS-mutant cancers in combination with other established cancer drugs or treatments such as radiation or chemotherapy might lead to effective therapeutics. Considering that a direct targeting of oncogenic KRAS remains elusive and challenging, an unbalanced redox state of highly malignant KRAS-mutant cancers is an attractive area for therapeutics development.

A high dose of VC is a powerful leverage in an induction of a glucose-dependent oxidative stress in KRAS-mutant cancer cells. It was demonstrated that absorption of the oxidized form of VC, DHA-resembling glucose, is sufficient to exhaust a cellular GSH system because of an intracellular reduction of DHA back to VC mediated by oxidation of GSH. It is evident that a high dose of VC is a powerful approach in inducing the oxidative stress selectively in KRAS-mutant cancer cells due to their enhanced glucose absorption. In view of the glucose-dependent impact of VC, a potent cytotoxic impact of VC in KRAS-mutant cancer cells is observed mostly at a low glucose cell culture condition. This means that VC alone, even at a high dose, is effective in inducing a potent oxidative stress in KRAS-mutant cancer cells and reaching a potent cytotoxic impact in clinical settings that might not be feasible consistently because of DHA’s competition with glucose.

Considering the selectivity of VC in inducing an oxidizing impact in KRAS-mutant cancer cells, there were several cancer clinical trials initiated to test a high dose of VC or its combination with other common cancer drugs such as gemcitabine, paclitaxel, cisplatin, or docetaxel, and in some cases including the radiation treatment [102]. Although the clinical efficacy of the high-dose VC applications are actively examined, it is also sensible to enhance an oxidizing effect of VC with another oxidizing drug, ATO. The powerful synergistic impact of the VC and ATO combination was observed to effectively kill selected KRAS-mutant cancer cells and was recently reported [161]. The study clearly indicates that a potentiating impact of ATO in enhancing the oxidizing effect of VC is escalated to a potent cytotoxic impact selectively in KRAS-mutant cancer cells without affecting primary and other cancer cells. Importantly, the ATO and VC combination was effective at high-glucose cell culture conditions, indicating a potency of the drug combination without interference of glucose presence that has been evident for the action of VC alone. The advantage of the ATO and VC combination was also evident in in vivo xenograft study, when VC was effective at a much lower (1.5 g/kg) dose with ATO when compared to the VC dose injection alone (4 g/kg). Moreover, a synergistic action of ATO and VC in killing KRAS-mutant cancer cells (the human SW620 and LOVO colorectal adenocarcinoma cell lines) has also been described by another group [162]. If a glucose uptake determines the effect of VC, it is likely that the oxidizing effect of an ATO and VC combination will be more effective under fasting conditions.

A key finding of the ATO- and VC-induced cytotoxicity was a mechanism of killing KRAS-mutant cancer cells (Figure 6c) [161]. It has been shown that a potent cytotoxic impact was executed by a robust mitochondrial production of ROS that can be designated as a suicidal ROS production by mitochondria (SRPM). How an oxidizing impact of two drugs triggers a robust production of ROS by mitochondria or inducing SRPM is a central question to be actively pursued. A synergistic cytotoxic impact induced by the ATO and VC combination indicates that surpassing an oxidative threshold in KRAS-mutant cancer cells affects the redox-sensitive cellular systems including the most vulnerable mitochondrial oxidative phosphorylation system (OXPHOS) complexes. Taking it into account, a synergy of the ATO and VC combination in killing KRAS-mutant cancer cells might take place by the two-step actions. In the first step, a depletion of GSH is mediated by a high dose of VC that disarms a cellular anti-oxidative shield. In the second step, ATO acts in a full capacity by attacking thiol-reactive groups on cellular proteins with a preference toward the components of mitochondrial OXPHOS complexes [163]. It is likely that a direct oxidation of thiol-reactive proteins within OXPHOS complexes leads to uncontrolled ROS production or SRPM. While the exact mechanism is yet to be defined, the ATO-induced ROS production by mitochondria is coherent with the observations of the arsenic-induced mitochondrial cardiotoxicity [164] or damage in cellular model [165]. Within the mitochondrial OXPHOS complexes, the iron-sulfur (Fe-S) clusters are essential cofactors mediating electron transfer within the mitochondrial respiratory chain. The critical cysteines within OXPHOS complexes are engaged directly by their thiol-reactive groups in a proper coordination of Fe-S clusters and electron transfer [138,166], which are likely the most sensitive thiol-reactive groups for the ATO binding and functional interference.

If this is the case, ATO can directly attack most critical (Fe-S) clusters of OXPHOS, collide electron transport system and trigger ROS generation at a high scale, resulting in cytotoxic impact by SRPM. Thus, according to the proposed hypothesis, the oxidation of thiol-reactive groups within the mitochondrial OXPHOS complexes by ATO is a critical step resulting in aberrant mitochondrial function and provoking SRPM in KRAS-mutant cancer cells, whereas a high dose VC action leads to a GSH depletion step that eliminates an interference with the ATO’s oxidation action (Figure 6c).

## 7. A Remarkable Turn: The VC’s Enantiomer *D*-VC Works Better in the Tumor Xenograft Animal Model

A cytotoxic action of ascorbates is mediated by a glucose-dependent oxidizing effect that determines a selectivity in killing of highly malignant KRAS-mutant cancer cells. Among several analogs of VC, only its enantiomer *D*-VC in combination with ATO showed a similar potent cytotoxic effect in killing KRAS-mutant cancer cells in cell culture [161]. It is coherent with the original study carried out in 1993 describing a direct cell-killing action of VC or *D*-VC on fast growing malignant cancer cells by defining their mechanism of action indicating that “certain oxidation and degradation products of ascorbate were cytotoxic agents” for malignant cancer cells [167]. *D*-VC, also known as erythorbic acid, is a common food additive because it is a non-toxic antioxidant. A lower toxicity of *D*-VC versus VC is expected because more rapid clearance of *D*-VC from human bodies has been reported [168]. In addition, a lower toxicity of *D*-VC is also supported by the higher tolerance and survival of mice injected with high doses of *D*-VC versus the VC injection effects (Begimbetova and Sarbassov, unpublished data). More detailed comparative toxicological studies of VC and *D*-VC will be highly informative in defining a chirality-dependent toxicity of VC.

An unexpected finding was made when the animal xenograft studies of VC and *D*-VC were carried side by side, although no difference was detected in their cytotoxic effects in the cell culture setting. The KRAS-mutant tumor xenograft study in mice showed a clear distinction in the tumor eradicating potency between the VC and *D*-VC enantiomers. The injections of ATO, VC or *D*-VC alone had only weak and partial tumor suppressing effects, but the ATO and *D*-VC combination had much more potent KRAS-mutant tumor suppressing impact compared to the group of tumor-bearing mice injected with ATO and VC [161]. This finding shows that the ATO and *D*-VC drug combination is superior to the ATO and VC injections in suppressing KRAS-mutant tumors only in mouse models that has not been detected in cell culture condition. It indicates that *D*-VC, a non-natural enantiomer of VC, has distinctive pharmacokinetic properties increasing its tumor-suppressing capacity in the animal model. It is a rare case when a stereoisomeric analog shows a higher potency in animal models that is highly relevant for enhancing clinical applications of VC.

There were no reports of clinical applications of *D*-VC that explains a scarceness of information on a chirality-dependent pharmacokinetics of VC. One possible explanation of a *D*-VC superiority in vivo might be related to the report indicating the eight times-slower oxidation rate of *D*-VC to DHA compared to the oxidation rate of its natural enantiomer VC [169]. It might lead to advantage for the *D*-VC treatment by resulting in a higher accumulation of its oxidized form DHA in circulation and, subsequently, inducing a more potent oxidizing impact in KRAS-mutant cancer cells.

## 8. Conclusions

A combination of two known drugs, ATO and VC, works synergistically by enhancing a glucose-dependent oxidation effect and selective killing of KRAS-mutant cancer cells. How an oxidative stress provoked by the ATO and VC treatment triggered a SRPM is an attractive mechanism to explore. Understanding of SRPM will explain how a potent cytotoxic impact occurs in KRAS-mutant cancer cells. A promising ATO and VC or ATO and *D*-VC drug combination has to be studied in clinical trials to evaluate efficacy of the drug combination in suppressing KRAS-mutant cancers. In the previous cancer clinical trials, ATO was administered only with a low dose of VC. To alleviate the toxic effect of ATO at its maximum concentration (0.25 mg/kg), the antioxidant oral supplement of VC (1 g/day) has been provided [170,171,172]. At low dose, VC was not effective to potentiate the action of ATO. It will be critical to carry out the cancer clinical trials by administering ATO with a high dose of VC. A chirality-dependent action of VC also has to be addressed by performing the pharmacokinetics of VC and *D*-VC that will determine why *D*-VC is more effective in animal models.

## Figures and Tables

**Figure 1 cells-11-03454-f001:**
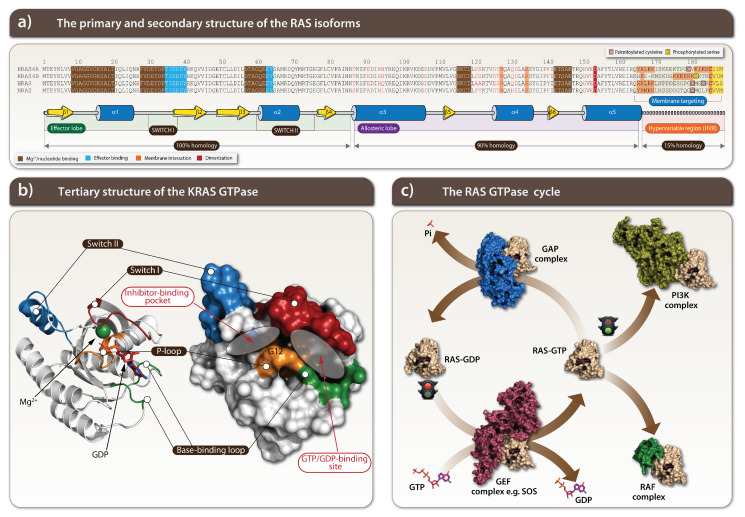
The human RAS small GTPase isoforms: HRAS, NRAS, KRAS4A and KRAS4B. (**a**) The human RAS isoforms share high homology in their primary and secondary structures. The multiple sequence alignment was done by CLUSTALW. The RAS G-domain has an effector lobe (100% homology), allosteric lobe (90% homology), and a hypervariable region (15% homology). The residues in the effector and allosteric lobes that do not share homology are highlighted in red. Important regions are shown in the sequence alignment as colored squares and are responsible for Mg^2+^/nucleotide-, effector-binding or membrane interaction and dimerization. The majority of residues in the hypervariable region are responsible for interaction of RAS isoforms with the cytosolic membrane and some of the cysteine and serine residues are post-translationally modified by palmitoylation or phosphorylation. The very last carboxy-terminal tetrapeptide CXXX motif undergoes isoprenylation. The KRAS GTPase contains five α-helices (blue cylinders) and six β-strands (yellow arrows). The location of functionally important regions such as switch I (residues 30–38) and II (residues 60–76) are highlighted under the schematic secondary structures. (**b**) The three-dimensional structure of the wild-type KRAS GTPase based on crystal structure (PDBID 4OBE) shows the location of the functionally important regions such as switch I (red) and II (blue), the P-loop (orange) responsible for the binding and correctly positioning the α and β phosphates of the nucleotides and the base-binding loop (green). The surface representation shows the GTP/GDP-binding site, the inhibitor-binding pocket and the location of the critical G12 residue located in the P-loop. (**c**) The RAS GTPase cycle shown on the example of the available HRAS crystal structure complexes. HRAS is shown in light yellow. The GTP-bound HRAS is activated (PDBIS 6Q21) and initiates cellular signaling by recruiting PI3K, a lipid kinase, (PDBID 1HE8) and/or RAF, a serine/threonine-specific protein kinase (PDBID 4G0N). The GAP complex (PBDID 1WQ1) accelerates the catalytic hydrolysis of GTP to GDP, where the subsequent GDP-bound RAS reaches an inactive state (PDBID 4Q21). The release of a GDP from the inactive RAS and the subsequent binding of GTP is stimulated by the GEF complex, containing SOS (PDBID 1NVV). Abbreviations: PI3K, phosphoinositide 3-kinase; RAF, rapidly accelerated fibrosarcoma; GAP, GTPase activating protein; GEF, guanine nucleotide exchange factor; SOS, son of sevenless homologue.

**Figure 2 cells-11-03454-f002:**
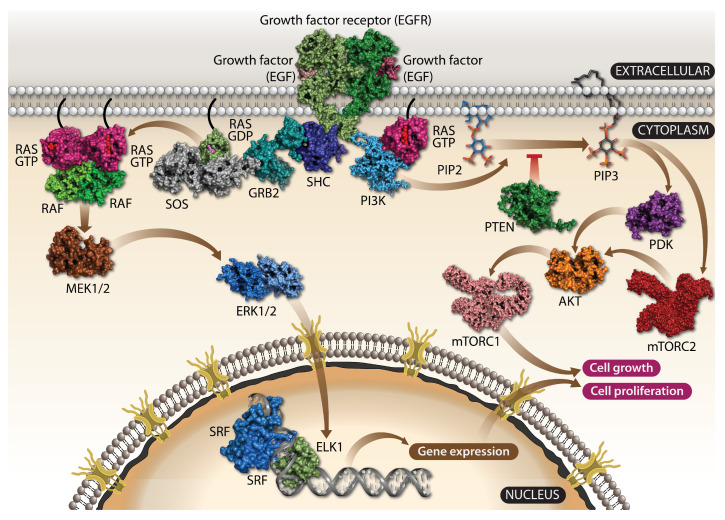
The RAF-MEK-ERK and PI3K-AKT-mTOR signalings are key RAS effector pathways. The RAS signaling, which is involved among other things in cell proliferation, apoptosis, migration, fate specification, and differentiation, is the prototypical kinase signaling pathway that starts with the binding of the growth factor such as EGF to its parent receptor, which is a receptor tyrosine kinase located in the cellular membrane. EGFR dimerizes, activates, and recruits SOS, a GEF factor, to its phosphorylated C-terminus through GRB2 and SHC. GEF catalyzes the GDP/GTP exchange, which in turn activates RAS. The activated GTP-bound RAS dimer associates with RAF, promoting its dimerization and subsequent activation. The phosphorylated RAF activates MEK1/2 followed by ERK1/2 activation. One of the targets of ERK1/2 is the transcription factor ELK1, which is phosphorylated as well. Activated ELK1 recruits its cofactor, a dimer of SRF, leading to the upregulation of target genes that direct cellular proliferation. The second pathway that is activated involves PI3K, which phosphorylates PIP2 to PIP3. This reaction may be reversed by PTEN. PIP3 subsequently interacts with PDK1, which phosphorylates AKT, a serine-threonine kinase and then finally the mTORC1 complex is assembled, which regulates cell growth. Alternatively, PIP3 influences the assembly of the mTOR2, which phosphorylates AKT. Abbreviations: EGFR, epidermal growth factor receptor; SOS, son of sevenless; GEF, guanine nucleotide exchange factor; SHC, SH2-adaptor protein; GRB2, growth factor receptor-bound protein 2; MEK1/2, mitogen-activated protein kinase kinase 1 and 2; SRF, serum response factor; PI3K, phosphatidylinositol 3-kinase; PIP2, phosphatidylinositol-4,5-bisphosphate; PIP3, phosphatidylinositol-3,4,5-trisphosphate; PTEN, phosphatase and tensin homologue; PDK1, phosphoinositide-dependent kinase-1; mTOR, mammalian target of rapamycin; MAPK, mitogen-activated protein kinase.

**Figure 3 cells-11-03454-f003:**
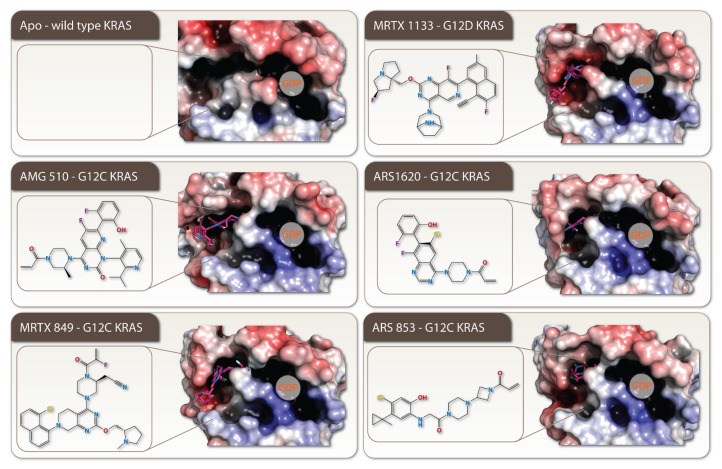
The recent advances in RAS inhibitor design. Five representative examples of recently development RAS inhibitors AMG 510 (PBDID 6OIM), MRTX 849 (PBDID 6OIM), MRTX 1133 (PBDID 7RPZ), ARS 1620 (PBDID 5V9U), and ARS 853 (PBDID 5F2E) are shown. On the left, the molecular structure of the inhibitors as well as their binding mode in the inhibitor-binding pocket is depicted. For simplicity, the GDP molecule is not displayed, but its position is indicated on the electrostatics surface representation of RAS. As a reference the wild-type apo RAS is also shown (PDBID 4OBE).

**Figure 4 cells-11-03454-f004:**
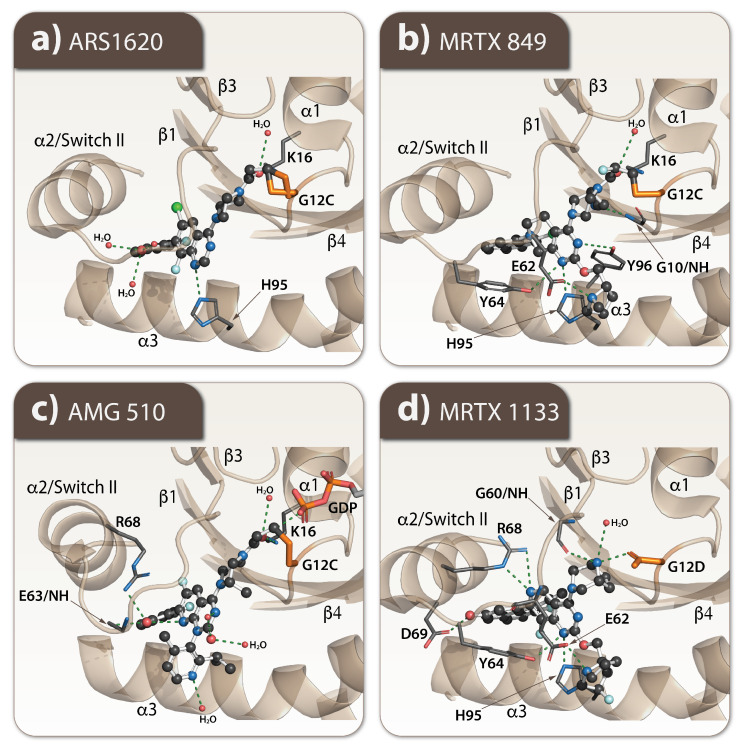
Mechanism of interaction of RAS inhibitors. The molecular interaction of selected inhibitors is shown. All the three G12C mutants have covalently bound ligands compared to G12D, which instead forms an H-bond with its D12. Moreover, all ligand-bound G12C mutants use K16 as a contact point; its position is well-conserved in all three structures. In the G12D structure, this interaction is missing. (**a**) ARS 1620 (PBDID 5V9U) interacts with K16 and H95 and uses three water-mediated contacts. (**b**) MRTX 849 (PBDID 5F2E) besides K16 and H95 also interacts with E62, Y64, and Y96, the backbone of G10, and one water molecule. In addition, it interacts with the GDP molecule that may prevent its exchange to GTP. (**c**) AMG 510 interacts with K16, backbone of E63, R68 from Switch II, and creates three water-mediated contacts. (**d**) MRTX 1133 also interacts with R68 from Switch II, D69, Y64, H95, E62, and the backbone of G60 and forms one water-mediated contact.

**Figure 5 cells-11-03454-f005:**
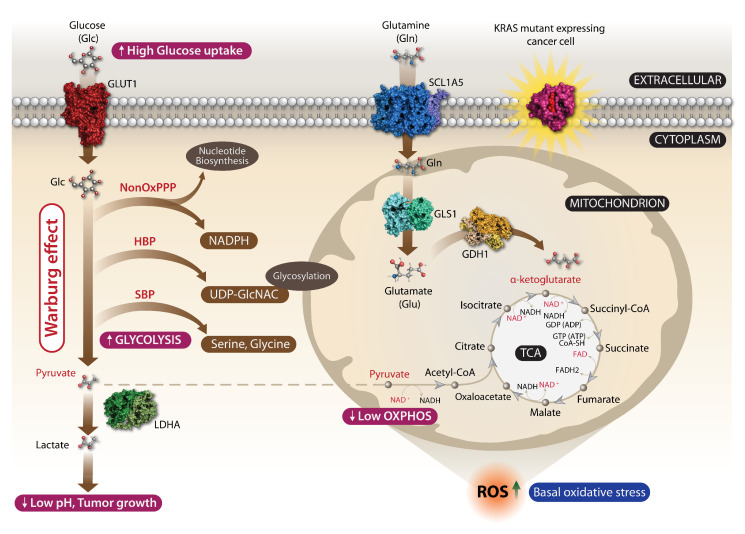
The Warburg effect in RAS-expressing cancer cells. RAS-expressing cells are exposed in the tumor microenvironment to hypoxia, acidosis, and stromal cell formation that significantly affect the glycolysis as well as important signaling such as PI3K and HIF1 (not displayed). All these lead to an additional impairment in the efficiency of the TCA cycle, where the OXPHOS is decreased. The major alteration comes from the choice of cancer cells to preferentially metabolize anaerobically glucose, even under normoxia, which is referred to as the Warburg effect. RAS-expressing cells show high glucose uptake by GLUT1, a major increase of the glycolysis and the massive production of lactate catalyzed by LDHA, from which only a very small amount enters the mitochondria. The majority of lactate is transported outside of the cell, further promoting low pH and tumor growth. In parallel with the decreased overall flux of pyruvate through the TCA cycle, there is an increase of the utilization of glutamine, in particular its carbon backbone and amino moiety either by entering the TCA cycle as α-ketoglutarate or transamination to sustain biosynthetic metabolic pathways such as amino acid, nucleotide, and glutathione synthesis. Glutamine is transported to the cell through the SCL1A5 transporter, entering the mitochondria where GLS1 hydrolysis converts it to glutamate and ammonia. Glutamate is then converted by oxidative deamination to α-ketoglutarate and ammonia catalyzed by GDH1 and enters the TCA cycle. Abbreviations: PI3K, phosphatidylinositol 3-kinase; HIF1, hypoxia-inducible factor-1; TCA cycle, tricarboxylic acid/Krebbs cycle; OXPHOS, oxidative phosphorylation; GLUT1, glucose transporter-1; LDHA, lactate dehydrogenase A; NonOxPPP, non-oxidative penthose phosphate pathway; HBP, hexosamine biosynthetic pathway; SBP, serine biosynthetic pathway; SCL1A5, alanine-serine-cysteine transporter 2; GLS1, K-type mitochondrial glutaminase (phosphate-activated amidohydrolase); GDH1, mitochondrial glutamate dehydrogenase.

**Figure 6 cells-11-03454-f006:**
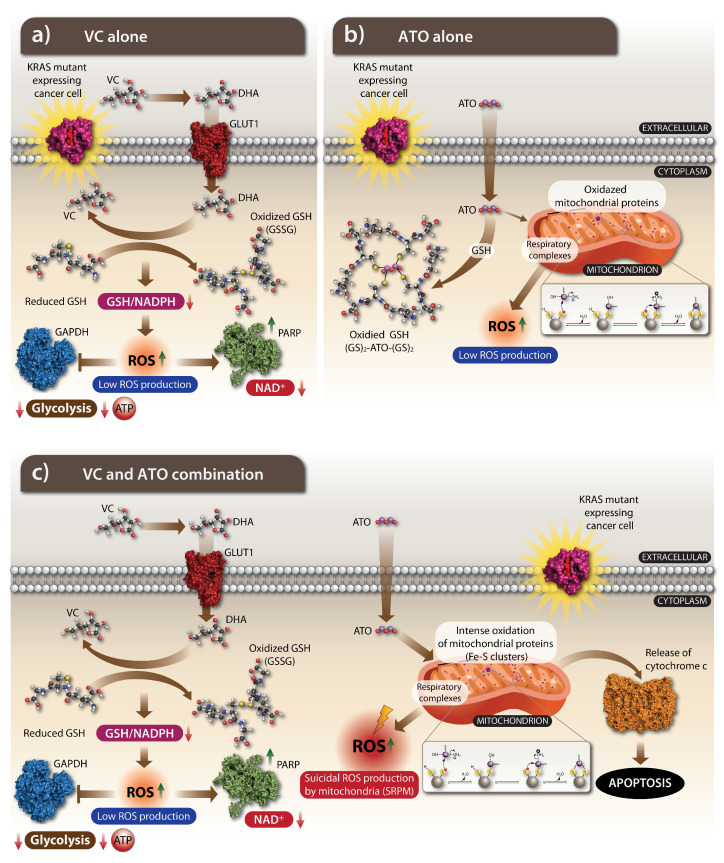
The cellular mechanism of VC, ATO and their co-treatment. (**a**) Increased uptake of DHA through GLUT1 causes oxidative stress as intracellular DHA is reduced to VC, depleting the pool of reduced GSH. This leads to oxidative stress with increased ROS production. The accumulation of ROS inactivates GAPDH, which in turn, in highly glycolytic KRAS-mutant cancer cells, leads to decreased glycolysis and ATP followed by an energetic crisis and subsequent cell deaths. In addition, the response to low GSH and NADPH and subsequent ROS production PARP is activated and NAD+ is decreased, leading to further inhibition of GADPH. (**b**) The main effect of ATO is to oxidize GSH so one molecule of ATO binds four molecules of oxidized GS (GS)2-ATO-(GS)2. A small portion of ATO enters the mitochondria, where it oxidizes the mitochondrial proteins, which leads to ROS production. (**c**) VC and ATO combination acts synergistically to enhance oxidative stress to cytotoxic effect. The VC depletes GSH by oxidizing it; thus ATO acts without much interference from GSH on the mitochondria by carrying out a direct intense oxidation of their Fe-S clusters, mostly located in OXPHOS complexes representing the respiratory electron transport chain. The heavy oxidation of the Fe-S clusters on OXPHOS impairs mitochondrial function triggering suicidal ROS production by mitochondria (SRPM) with subsequent release of cytochrome C from mitochondria and initiation of the apoptotic pathway and cell death. Abbreviations: GSH, glutathione (reduced); GS, glutathione (oxidized); ROS, reactive oxygen species; GAPDH, glyceraldehyde 3-phosphate dehydrogenase; PARP, poly ADP-ribose polymerase; SRPM, suicidal ROS production by mitochondria.

**Table 1 cells-11-03454-t001:** Recent advances in targeting specific KRAS mutations.

Name	Marker	Reference
AMG510/Sotorasib (Amgen)	KRAS G12C	[37]
MRTX849/Adagrasib (Mirati)	KRAS G12C	[38]
D-1553 (Iventis Bio)	KRAS G12C	[39,40]
JDQ443 (Novartis)	KRAS G12C	[41,42]
MRTX1133 (Mirati)	KRAS G12D	[43]
JNJ-74699157 (Johnson & Johnson)	KRAS G12C	[44]
LY3499446 (Eli Lilly & Company)	KRAS G12C	[45] NCT04165031 discontinued
iExosomes	KRAS G12D	[46]
Anti-KRAS G12D mTCR PBL	KRAS G12D	[47]
Anti-KRAS G12V mTCR PBL	KRAS G12V	[47]

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
