# Peer review of "Enhancing an Oxidative “Trojan Horse” Action of Vitamin C with Arsenic Trioxide for Effective Suppression of KRAS-Mutant Cancers: A Promising Path at the Bedside"

_cells, 2022, doi:10.3390/cells11213454_

Round 1

Reviewer 1 Report

In the review the authors take into consideration the role of mutant KRAS in tumour development. it is well known that KRAS mutations are very frequent in cancer and a lot of tumours become addicted to KRAS. The effects of mutations are important for changes in metabolism and are crucial for driving cancer development. Moreover the biggest problem is represented by the undruggability of KRAS-mutaded cancers: in this types of tumours it's possible to face with metabolic reprogramming which fuel cancer progression. The biggest challenge is how to specifically and usefully target KRAS gene. Even if Vitamin C represent a good candidate to treat KRAS mutant cancer cells, and its combination with arsenic trioxide (ATO) might be a promising drug combination to killing selectively KRAS mutant cancer cells, a lot of work must be done to decrease the side effects related to ATO. 

Very good work.

Specific comments:

- line 114: please check the phrase

- line 283: H2O2 need to be corrected

- line 293-297: the phrase is too long, please check and riformulate

- line 302: correct NOX I with NOX1

- lines 376-380: it seems a ripetition of thing just said in the previous lines

- lines 398-400: check the phrase, maybe there is a verb missed

- line 423: there are two commas

Author Response

Reviewer 1

In the review the authors take into consideration the role of mutant KRAS in tumour development. it is well known that KRAS mutations are very frequent in cancer and a lot of tumours become addicted to KRAS. The effects of mutations are important for changes in metabolism and are crucial for driving cancer development. Moreover the biggest problem is represented by the undruggability of KRAS-mutaded cancers: in this types of tumours it's possible to face with metabolic reprogramming which fuel cancer progression. The biggest challenge is how to specifically and usefully target KRAS gene. Even if Vitamin C represent a good candidate to treat KRAS mutant cancer cells, and its combination with arsenic trioxide (ATO) might be a promising drug combination to killing selectively KRAS mutant cancer cells, a lot of work must be done to decrease the side effects related to ATO. 

Very good work.

  1. the line 114: please check the phrase

In silico drug discovery methods employing drug--gene network analysis by screened the FDA approved drug library that potentially can work as inhibitors of oncogenic KRAS which further could be repurposed. 

HAS BEEN CHANGED TO

In silico drug discovery methods employing drug--gene network analysis, that screen the FDA approved drug library, can lead to repurposed inhibitors of oncogenic KRAS.

2. the line 283: H2O2 need to be corrected

HAS BEEN CORRECTED

3. the line 293-297: the phrase is too long, please check and reformulate

Expression of K-ras G12V protein caused dysfunction of mitochondria by significant decrease in mitochondrial respiratory activity (complex I) and decrease in oxygen consumption, decrease of mitochondrial membrane potential âˆ†Ψm, elevated ROS generation and adaptive upregulation of GSH synthesis in response to the sustained ROS production associated with KRAS G12V [86].

HAS BEEN CHANGED TO 

Expression of KRAS G12V protein causes dysfunction of mitochondria by significant decrease in mitochondrial respiratory activity (complex I), oxygen consumption and mitochondrial membrane potential ∆Ψm. In addition, elevated ROS generation and adaptive upregulation of GSH synthesis in response to the sustained ROS production have been associated with KRAS G12V mutant [86]. 

  1. the line 302: correct NOX I with NOX1

CORRECTED

5. the lines 376-380: it seems a repetition of thing just said in the previous lines

A high dose of VC kills cancer cells by provoking an oxidative stress in highly malignant cancer cells including KRAS mutant cancer cells. How does a potent antioxidant VC become an oxidative factor in cancer cells? The oxidative impact is carried out by the oxidized form of VC known as dehydroascorbate (DHA) and the distinct locations of the oxidation VC to DHA reaction and reducing DHA back to VC reaction determine the oxidation of cancer cells. VC gets oxidized in a body to DHA. KRAS mutant cancer cells actively absorb DHA by a glucose transporter GLUT1 because DHA resembles glucose. DHA inside a cancer cell is reduced back to VC by the cost of glutathione oxidation. In a high scale, an active absorption of DHA, a decoyed form of glucose, results in a substantial exhaustion of a cellular GSH leading to oxidative stress coupled to ROS accumulation and inactivation of glyceraldehyde 3-phosphate dehydrogenase (GAPDH). A continuous inhibition of GAPDH activity is cytotoxic for highly glycolytic KRAS mutant cancer cells \citep{Yun2015, Ngo2019}.

Metabolic reprogramming sensitizes KRAS mutant cancer cells to a high dose of VC by meddling in an epigenetic regulation, hypoxia signaling and intracellular iron metabolism (Figure~\ref{fig6}a) \citep{Kaźmierczak-BaraÅ„ska2020}. It becomes evident that a pro-oxidant action of VC depends on metal-ion redox chemistry. In particular, free iron was shown to be essential for VC-induced cytotoxicity \citep{Chen2007}. The oxidized form of VC, DHA is an active molecule with short half-life and structure similar to glucose, and thus can be transported via GLUTs to cells. KRAS metabolic reprogramming associated with elevated expression of GLUT1 allows for increased uptake of DHA to the cytoplasm which leads to a disruption in the Warburg effect \citep{Chen2007, Aguilera2016}. Inside the cell DHA is further reduced to VC by GSH, NADH and other enzymes and induces oxidative damage via auto-oxidation.

HAS BEEN CHANGED TO

Metabolic reprogramming sensitizes KRAS mutant cancer cells to a high dose of VC by meddling in an epigenetic regulation, hypoxia signaling and intracellular iron metabolism (Figure~\ref{fig6}a) \citep{Kaźmierczak-BaraÅ„ska2020}. The high dose of VC kills cancer cells by provoking an oxidative stress in highly malignant cancer cells including KRAS mutant cancer cells, but how does a potent antioxidant VC become an oxidative factor in cancer cells? It is highly evident that a pro-oxidant action of VC depends on metal-ion redox chemistry. In particular, free iron was shown to be essential for VC-induced cytotoxicity \citep{Chen2007}. The oxidative impact is carried out by the oxidized form of VC known as dehydroascorbate (DHA) and the distinct locations of the oxidation VC to DHA reaction and reducing DHA back to VC reaction determine the oxidation of cancer cells. VC gets oxidized in a body to DHA. KRAS mutant cancer cells actively absorb DHA by a glucose transporter GLUT1 because DHA resembles glucose. KRAS metabolic reprogramming associated with elevated expression of GLUT1 allows for increased uptake of DHA to the cytoplasm which leads to a disruption in the Warburg effect \citep{Chen2007, Aguilera2016}. DHA inside a cancer cell is reduced back to VC by the cost of GSH oxidation. In a high scale, an active absorption of DHA, a decoyed form of glucose, results in a substantial exhaustion of a cellular GSH leading to oxidative stress coupled to ROS accumulation and inactivation of glyceraldehyde 3-phosphate dehydrogenase (GAPDH). A continuous inhibition of GAPDH activity induces oxidative damage via auto-oxidation, which is cytotoxic for highly glycolytic KRAS mutant cancer cells \citep{Yun2015, Ngo2019}.

  1. the lines 398-400: check the phrase, maybe there is a verb missed

Besides, it has been observed that downregulation of PDK-1 by VC mediated by hydroxylation of proline (P402) in a hypoxia inducible factor-1α (HIF-1α) [117].

HAS BEEN CHANGED TO

Besides, the downregulation of PDK-1 has been linked to a VC-mediated hydroxylation of proline (P402) in a hypoxia inducible factor-1α (HIF-1α) [117].

7. line 423: there are two commas

CORRECTED

Sincerely,

Dos Sarbassov. 

Reviewer 2 Report

This is a very comprehensive review of an important field with some very interesting studies related to the additive effect of vitamin c and arsenic trioxide.

The state of KRAS druggability is very well reviewed. The figures are very well done. The text is mostly easy to read. The references are comprehensive and  adequate. The data on Vit C and ATO are compelling. The mechanisms underlying the potent effect of this combination is well discussed.  

I have only one minor comment that concerns the abstract which should be edited to make it clearer.

Author Response

A point-by-point response to the reviewer’s comments:

Reviewer 2.

This is a very comprehensive review of an important field with some very interesting studies related to the additive effect of vitamin c and arsenic trioxide.

The state of KRAS druggability is very well reviewed. The figures are very well done. The text is mostly easy to read. The references are comprehensive and adequate. The data on Vit C and ATO are compelling. The mechanisms underlying the potent effect of this combination is well discussed.  

I have only one minor comment that concerns the abstract which should be edited to make it clearer.

As recommended, we edited the Abstract accordingly to make it more clear by emphasizing the drug combination application.

Original Abstract

The turn-on mutations of KRAS gene, coding a small GTPase coupling growth factor signaling, are contributing to nearly 25% of all human cancers leading to highly malignant tumors with poor outcome. Due to lack of distinctive pockets the druggability of KRAS is challenging yet recently, the specific G12C mutant KRAS inhibitors have been developed with high relevance to clinical outcome. However, drug resistance has been acquired as a result of the additional KRAS mutations interfering with the drug-binding. To address this, a limited monotherapy approach can be further explored with developing novel therapeutic drug combinations to target metabolic breaches typical for cancer cells. One of them is a high glucose consumption leading to unbalanced redox state and sensitivity to oxidative stress. Here, we discuss the use of an unexpected candidate vitamin C (VC) acting in high dose as an oxidative "Trojan horse" for KRAS mutant cancer cells that can be potentiated with another oxidizing drug arsenic trioxide (ATO) to obtain a potent cytotoxic impact. Moreover, we outline the advantages of VC’s non-natural enantiomer D-VC because of its distinctive pharmacokinetics and lower toxicity. Thus, the D-VC/ATO combination shows a promising path to treat KRAS mutant cancers in clinical settings.

EDITED ABSTRACT

Abstract: The turn-on mutations of KRAS gene, coding a small GTPase coupling growth factor signaling, are contributing to nearly 25% of all human cancers leading to highly malignant tumors with poor outcome. Targeting of oncogenic KRAS remains as a most challenging task in oncology. Recently, the specific G12C mutant KRAS inhibitors have been developed but with a limited clinical outcome because of acquiring a drug resistance. Alternatively, exploiting of a metabolic breach of KRAS mutant cancer cells related to a glucose-dependent sensitivity to oxidative stress is becoming a promising indirect cancer targeting approach. Here, we discuss the use of a vitamin C (VC) acting in high dose as an oxidative "Trojan horse" agent for KRAS mutant cancer cells that can be potentiated with another oxidizing drug arsenic trioxide (ATO) to obtain a potent and selective cytotoxic impact. Moreover, we outline the advantages of VC’s non-natural enantiomer D-VC because of its distinctive pharmacokinetics and lower toxicity. Thus, the D-VC and ATO combination shows a promising path to treat KRAS mutant cancers in clinical settings.

We will be happy to provide an additional information upon your request

We look forward to hearing from you.

Sincerely,

Dos Sarbassov. 
